# Pathways Affected by Falcarinol-Type Polyacetylenes and Implications for Their Anti-Inflammatory Function and Potential in Cancer Chemoprevention

**DOI:** 10.3390/foods12061192

**Published:** 2023-03-11

**Authors:** Ruyuf Alfurayhi, Lei Huang, Kirsten Brandt

**Affiliations:** 1Human Nutrition & Exercise Research Centre, Population Health Sciences Institute, Faculty of Medical Sciences, Newcastle University, Newcastle upon Tyne NE1 7RU, UK; 2Department of Food Science and Human Nutrition, College of Agriculture and Veterinary Medicine, Qassim University, Qassim, Buraydah 52571, Saudi Arabia; 3Immunity and Inflammation Research Theme, Translational and Clinical Research Institute, Faculty of Medical Sciences, Newcastle University, Newcastle upon Tyne NE1 7RU, UK

**Keywords:** polyacetylenes, phytochemicals, anti-inflammatory, anticancer

## Abstract

Polyacetylene phytochemicals are emerging as potentially responsible for the chemoprotective effects of consuming apiaceous vegetables. There is some evidence suggesting that polyacetylenes (PAs) impact carcinogenesis by influencing a wide variety of signalling pathways, which are important in regulating inflammation, apoptosis, cell cycle regulation, etc. Studies have shown a correlation between human dietary intake of PA-rich vegetables with a reduced risk of inflammation and cancer. PA supplementation can influence cell growth, gene expression and immunological responses, and has been shown to reduce the tumour number in rat and mouse models. Cancer chemoprevention by dietary PAs involves several mechanisms, including effects on inflammatory cytokines, the NF-κB pathway, antioxidant response elements, unfolded protein response (UPR) pathway, growth factor signalling, cell cycle progression and apoptosis. This review summarises the published research on falcarinol-type PA compounds and their mechanisms of action regarding cancer chemoprevention and also identifies some gaps in our current understanding of the health benefits of these PAs.

## 1. Introduction

Studies have indicated the beneficial impact of eating vegetables and fruits on human health in preventing chronic diseases including cancer, which is one of the major causes of death around the world [1]. Polyacetylenes (PAs) are a class of chemicals defined by the presence of two or more carbon–carbon triple bonds in the carbonic skeleton [2]. Falcarinol-type PAs are biologically active compounds that are widely found in plants in the Apiaceae family, such as carrots, celery and parsley, and the Araliaceae family, such as ginseng. Carrot is the main dietary source of polyacetylenic oxylipins, including falcarinol (FaOH), falcarindiol (FaDOH) and falcarindiol 3-acetate (FaDOH3Ac) (Figure 1), with FaOH serving as the intermediate metabolite of PA, from which the other forms are generated [3,4,5]. Carrots have been studied for their nutritional value, in addition to their disease-curative effects, for almost 90 years [6]. Carrot is a rich source of the vitamin A precursor β-carotene and also provides some potentially beneficial dietary fibre. Carrot also contains other potentially bioactive phytochemicals including carotenoids, phenolics, PAs, isocoumarins, terpenes and sesquiterpenes, many of which have been extensively investigated for potential therapeutic properties against a wide range of diseases including cancer, cardiovascular disease, diabetes, anaemia, colitis, ocular diseases and obesity [7]. Ginseng is also rich in PAs; in addition to FaOH (also called panaxynol), they include panaxydiol and panaxydol (Figure 1) [8], which have similar properties to FaOH [9,10,11]. Despite the extensive research on the analytical and biochemical identification and characterization of plant PAs, as well as the large number of papers on their putative biological functions, little is known about the structures and functions of the enzymes involved in PA biosynthesis [12]. Furthermore, the molecular genetic principles underlying PA production in various plant tissues are poorly understood, and little is known about the genetics and inheritance of specific PA patterns and concentrations in (crop) plants [3].

β-carotene was initially believed to be protective against multiple chronic diseases, particularly cardiovascular disease and cancer, due to observations of associations of reduced risk of these diseases with a high dietary intake (with carrots as the primary dietary source of β-carotene) [13,14,15,16]. However, meta-analyses of subsequent intervention studies ruled out a role of β-carotene in suppressing non-communicable diseases affecting lifespan [17]. for example, dietary supplementation with purified β-carotene showed a dose-dependent increased risk of lung cancer for intakes higher than what can be obtained from food [18]. Thus, it was suggested that other bioactive substances such as PAs (FaOH and FaDOH) could be responsible for the health benefits of carrot [19]. Phytochemicals have been used as the main sources of the primary structures for conventional drugs used for curing cancer. Natural products historically have been essential in the development of new treatments for cancer and infectious diseases [20,21,22,23]. This review focuses on the possible mechanisms of action of PAs in inflammation and cancer, with emphasis placed on the various pathways involved including growth factor signalling, inflammatory processes, oxidative stress, cell cycle progression and apoptosis. We will also discuss the potential for PAs to modulate one or more of these pathways to contribute toward the treatment or prevention of inflammation and cancers.

## 2. Polyacetylenes and Inflammation

### 2.1. Chronic Inflammation Disease and Cancer

Chronic inflammation is recognized as a leading promoting factor of diseases including carcinogenesis [24], which continues to be the leading cause of mortality and disability around the world [25,26,27,28]. Tumour-promoting inflammation is recognised as an enabling hallmark of cancer [29]. Cancer and inflammation are linked by intrinsic and extrinsic pathways. Intrinsically, oncogenes regulate the inflammatory microenvironment, whereas extrinsically, the inflammatory microenvironment promotes the growth and spread of cancer [30]. Various cell types involved in chronic inflammation can be found in tumours, both in the surrounding stroma and within the tumour itself. Neoplasms, including some of epithelial origin, contain a significant inflammatory cell component [31]. Multiple studies on human clinical samples have revealed that inflammation influences epithelial cell turnover [32,33]. Significantly, human susceptibility to breast, liver, large bowel, bladder, prostate, gastric mucosa, ovary and skin carcinoma is increased when proliferation occurs in the context of chronic inflammation [32,33,34,35,36,37].

Chronic inflammation is linked to approximately 25% of all human cancers and increases cancer risk [38] by stimulating angiogenesis and cell proliferation, inducing gene mutations and/or inhibiting apoptosis [38]. Chronic inflammation can develop from acute inflammation if the irritant persists, although in most cases the response is chronic from the start. Chronic inflammation is characterized by the infiltration of injured tissue by mononuclear cells such as macrophages, lymphocytes and plasma cells, as well as tissue destruction and attempts at repair [31]. *Helicobacter pylori* infections in gastric cancer, human papillomavirus infections in cervical cancer, hepatitis B or C infections in hepatocellular carcinoma and inflammatory bowel disease in colorectal cancer (CRC) are common causes of chronic inflammation associated with cancer development [39,40]. Inflammation also causes epigenetic changes that are linked to cancer development.

Natural PAs from diverse food and medicinal plants and their derivatives exert multiple bioactivities, including anti-inflammatory properties [41]. PAs can impact inflammation through known and unknown pathways. Evidence supports that PA compounds improve human health by stimulating anti-cancer and anti-inflammatory mechanisms [3]. These PAs contain triple bonds that functionality convert them into highly alkylating compounds that are reactive to proteins and other biomolecules. This unique molecular structure might be the key to understanding the beneficial effects of PAs such as their anti-inflammatory and cytoprotective function [41]. Recent research has suggested that the anti-cancer role of certain foods might be attributed to their anti-inflammatory function. Root vegetables, and particularly carrots, are promising natural sources in this respect thanks to their rich content of PAs [3,41,42,43]. The anti-inflammatory properties of purple carrots have been suggested to be due to the high levels of anthocyanin pigments [44]; however, another study showed that PAs, not anthocyanins, are responsible for the anti-inflammatory bioactivity of purple carrots [45]. In vitro and in vivo studies have demonstrated that the health-benefitting effects of carrots and other root vegetables might be attributed to PAs, such as FaOH and FaDOH [46]. Other dietary compounds, including several different phytochemicals, have been examined in the context of cancer chemoprevention; however, until now the measured effects [47,48,49,50] have been quite small and inconsistent compared with those found for PAs.

### 2.2. Inhibition of Nuclear Factor Kappa B (NF-κB) Pathways

NF-κB is a transcription factor that regulates the expression of many genes involved in the regulation of inflammation and autoimmune diseases [51,52]. Moreover, NF-κB plays a significant role in inflammation-induced cancers, as NF-κB is one of the major inflammatory pathways that are triggered by, for example, infections causing chronic inflammation [39,40,53]. Cellular immunity, inflammation and stress are all regulated by NF-κB signalling, as are cell differentiation, proliferation and apoptosis (Figure 2) [54,55]. Both solid and hematologic malignancies frequently modify the NF-κB pathway in ways that promote tumour cell proliferation and survival [56,57,58].

NF-κB, a key factor in the inflammatory process, provides a mechanistic link between inflammation and cancer, and the components of this pathway are targets for chemoprevention, particularly in CRC [59]. There are two major signalling pathways for NF-κB activation, namely the canonical and the non-canonical NF-κB signalling pathways. The canonical pathway activates NF-κB1 p50, RELA and c-REL, which are also called canonical NF-κB family members. The non-canonical NF-κB pathway, on the other hand, selectively activates p100-sequestered NF-κB members, mostly NF-κB2 p52 and RELB, which are also called non-canonical NF-κB family members [60]. LPS and proinflammatory cytokines, among other pathogenic substances, activate NF-κB through degrading inhibitors of κB (IκBs) [61] to release the common subunit P65 (RELA). In order to trigger the transcription of these genes, activated NF-κB travels into the nucleus and attaches to its associated DNA motifs. When activated, the NF-κB p65 subunit binds to the promoter regions of genes involved in inflammation, leading to the production of *IL-6*, *IL-1β* and *TNF-α* [62].

Carrot PAs, particularly FaOH and FaDOH, were studied for their anti-inflammatory properties [3,63], in part by inhibiting the transcription factor NF-κB [64]; however, their exact mechanism of action is still unknown. Mice fed a diet containing FaOH were less likely to develop severe inflammation after being exposed to LPS [5]. FaOH from *Saposhnikovia divaricata (S. divaricata)* significantly reduced the levels of LPS-induced *TNF-α* and *IL-6* in cultured BV-2 microglia cells and murine serum [61]. FaOH and FaDOH purified from carrots were demonstrated for their preventative effects on colorectal precancerous lesions in azoxymethane (AOM)-induced rats. Biopsies of neoplastic tissue were analysed for gene expression, and the results showed that FaOH and FaDOH inhibited NF-κB and the downstream inflammatory markers *TNF-α, IL-6 and COX-2* [46]. FaOH from Radix Saposhnikoviae (dried roots of *S. divaricata*, Apiaceae) inhibited LPS-induced NF-κB p65 activation and IκB-α phosphorylation in BV-2 microglia cells [61]. Treatment using FaOH from the roots of *Heracleum moellendorffii* (*H. moellendorffii*) inhibited LPS-induced NF-κB signalling activation by inhibiting IκB-α degradation and nuclear accumulation of p65 [65] in RAW264.7 cells. In addition, FaDOH reduced the level of LPS/IFNγ-induced NF-κB, IKK-α and IKK-β activation in rat primary astrocytes [64].

Prostaglandin (PG) synthesis is a hallmark of inflammation. Two enzymes, cyclooxygenase (COX) 1 and 2, catalyse the first step of PG synthesis, but *COX-2* is the major one that responds to inflammatory signals to produce PG at inflammatory sites [66]. However, *COX-2* can be suppressed by inhibiting the NF-κB translocation pathway (Figure 2) [67]. *COX-2* expressions in healthy tissues are low, but they can quickly increase in response to growth factors, cytokines and signals promoting tumour invasion, metastasis, aberrant proliferation and angiogenesis [68]. Malignancies, including colorectal [69], bladder [70], breast [71], lung [72], pancreatic [73], prostate [73] and head and neck cancer [74], tend to be associated with elevated levels of *COX-2*. Mechanistically, *COX-2* promotes carcinogenesis through the creation of prostaglandins (PGs), which suppress apoptosis and stimulate the development of blood vessels in tumour tissue, which helps in sustaining tumour cell viability and growth [39,75], suggesting that anti-inflammatory drugs targeting *COX-2* might be beneficial in the treatment of different types of cancer.

PAs modulate inflammation via suppressing *COX-2* expression, which depends on NF-κB activation by inflammation [76]. FaOH inhibited LPS-induced *COX-2* expression in RAW264.7 cells, thus blocking PGE2 overproduction [65]. FaOH isolated from American ginseng (*P. quinquefolius*) effectively reduced the severity of colitis in mice treated by dextran sulphate sodium (DSS) induced for a week before FaOH treatment. FaOH reduced the number of CD11b+ macrophages in the lamina propria and the inflammation hallmark protein *COX-2*. These data suggest that macrophages expressing *COX-2* might be an essential factor for colitis development. Interestingly, FaOH treatment prior to DSS did not prevent colitis or reduce colitis severity in mice [8]. Quiescent macrophages in the lamina propria in a healthy mouse might offer protection against colitis induction, as depleting macrophages prior to induction of colitis may exacerbate DSS-induced colitis [77]. However, when colitis develops, there is an increase in the number of activated macrophages that secrete pro-inflammatory cytokines to boost the inflammatory response, thus exacerbating colitis. At this stage, an overactive macrophage response to enteric microbiota greatly contributes to the pathogenesis of colitis [78]. Treatment with FaOH to target macrophages was shown to be highly effective in suppressing colitis at this stage, highlighting the utility of FaOH in the treatment of a hyper-inflammatory disease (Figure 3) [79]. In an azoxymethane (AOM)-induced rat colorectal cancer model, FaOH and FaDOH downregulated *COX-2* in precancerous lesions of CRC [46] and also reduced the number of malignant tumour foci.

### 2.3. Oxidative Stress

#### 2.3.1. Inhibition of Nitric Oxide Synthase (NOS) and Pro-Inflammatory Cytokine Pathways

Nitric oxide (NO) is essential in a number of physiological functions, such as host defence, where it prevents the spread of disease-causing microbes within cells by stifling their reproduction [80]. The upregulation of NO expression in response to cytokines or pathogen-derived chemicals is a crucial part of the host’s defence against different types of intracellular pathogens. Different cell types produce the enzyme NOS, which catalyses the synthesis of NO, at high levels in a number of different tumour types [81]. Inflammation induces a specific form of NOS, i.e., the inducible isoform of nitric oxide synthase (iNOS), via activating *iNOS* gene transcription (Figure 2) [82]. iNOS is involved in complex immunomodulatory and antitumor mechanisms, which have a role in eliminating bacteria, viruses and parasites [83].

A considerable number of studies have been published on the role of PAs in *iNOS* expression in inflammation. Studies have demonstrated that FaOH extracted from *P. quinquefolius* inhibited *iNOS* expression in ANA-1 mΦ macrophage cells that were polarized to M1 [8] and LPS-induced iNOS expression in macrophages [84,85], leading to colitis suppression [8]. Moreover, FaDOH was tested on rat primary astrocytes for its impact on LPS/IFN-γ-induced *iNOS* expression. FaDOH blocked 80% of LPS/IFN-γ-induced *iNOS* by reducing *iNOS* protein and mRNA in a dose-dependent manner. FaDOH was shown to suppress *iNOS* expression, and it inhibited *IKK, JAK, NF-κB* and *Stat1* (Figure 2 and Figure 3) [64].

Another study showed a dose-dependent reduction in nitric oxide production in macrophage cells, where treatment with an extract of purple carrots containing PAs significantly reduced iNOS activity and *iNOS* expression in macrophage cells [45]. PAs reduced nitric oxide production in macrophage cells without cytotoxicity [45]. In vivo, purple carrots also inhibited inflammation in colitic mice and reduced colonic mRNA expression of *iNOS* [44]. FaOH from *H. moellendorffii* roots inhibits the LPS-induced overexpression of *iNOS* in RAW264.7 cells [65]. FaOH and other PAs from *P. quinquefolius* such as panaxydiol have a suppressive effect on *iNOS* expression in macrophages treated with LPS [85].

#### 2.3.2. Reactive Oxygen Species (ROS) Pathways

Oxidative stress is described as an imbalance between the generation of free radicals and reactive metabolites, also known as oxidants or reactive oxygen species (ROS), and their removal by protective mechanisms, also known as antioxidants. Electron transfer is involved in oxidative and antioxidative processes, which influence the redox state of cells and the organism. The altered redox state stimulates or inhibits the activities of various signal proteins, which have an effect on cell fate [86,87]. PAs promote apoptosis preferentially in cancer cells mediated ROS stress. A study has analysed ROS production in MCF-7 cells after treating with panaxydol. The increase in the ROS levels started at 10–20 min after the panaxydol treatment. The role of NADPH oxidase was investigated in order to determine the source of ROS after panaxydol treatment. The creation of reactive oxygen species (ROS) by NADPH oxidase appeared to take precedence, while ROS production in the mitochondria was secondary but also necessary, suggesting that NADPH oxidase generates ROS in the presence of panaxydol. Panaxydol was tested on different cell lines to investigate whether the induction of apoptosis occurred preferentially in cancer cells. In this study, panaxydol induced apoptosis only in cancer cells [88].

FaOH and FaDOH from carrot were tested for their effects on the oxidative stress responses of primary myotube cultures. The effects of 100 μM of H_2_O_2_ on the intracellular formation of ROS, the transcription of the antioxidative enzyme, cytosolic glutathione peroxidase (cGPx), and the heat shock proteins (HSP) HSP70 and heme oxygenase 1 (*HO-1*) were studied after 24 h treatment with FaOH and FaDOH at a wide range of concentrations. At intermediate concentrations, under which only moderate cytotoxicity was shown, intracellular ROS formation was slightly enhanced by PAs. In addition, PAs increased the transcription of cGPx and decreased the transcription of HSP70 and *HO-1*. The enhanced cGPx transcription may have decreased the need for the protective properties of HSPs as an adaptive response to the elevated ROS production. However, ROS production was significantly reduced with higher doses of PAs (causing substantial cytoxicity), and the transcription of HSP70 and *HO-1* decreased to a lesser extent, while the induction of cGPx was marginally reduced, showing a necessity for the protective and repairing functions of HSPs [89].

Transcription factor *Nrf2* (also known as nuclear factor erythroid 2-like 2) regulates the expression of various antioxidant, anti-inflammatory and cytoprotective factors, including heme oxygenase-1 (*Hmox1, HO-1*) and NADPH:quinone oxidoreductase-1 (*NQO1*). S-alkylation of the protein Keap1, which normally inhibits *Nrf2*, is induced by FaDOH extracted from *Notopterygium incisum* (*N. incisum*), as reported in [90]. Moreover, nuclear accumulation of *Nrf2* and expression of *HO-1* were both enhanced in LPS-stimulated RAW264.7 cells by FaOH from *H. moellendorffii* roots [65]. FaOH also inhibited the inflammatory factor level and reduced nitric oxide production in BV-2 microglia. FaOH also reduced the levels of LPS-induced oxidative stress in BV-2 microglia [61]. In addition, FaOH inhibited inflammation in macrophages by activating *Nrf2* [85]. HO-1 is linked to redox-regulated gene expression. Chemical and physical stimuli that produce ROS either directly or indirectly cause the expression of *HO-1* to respond [91]. A one-week study looked at the effects of FaOH (5 mg/kg twice per day in CB57BL/6 mice) pre-treatment on acute intestinal and systemic inflammation. FaOH effectively increased *HO-1* mRNA and protein levels in both the mouse liver and intestine and reduced the levels of the plasma chemokine eotaxin and the myeloid inflammatory cell growth factor GM-CSF, both of which are involved in the recruitment and maintenance of first-responder immune cells [92].

## 3. Unfolded Protein Response (UPR) Pathways

The endoplasmic reticulum (ER) stress response, also known as unfolded protein response (UPR), is a cellular process that is activated by a number of conditions that disrupt protein folding in the ER. The UPR is an evolutionarily conserved adaptive mechanism in eukaryotic cells that aims to clear unfolded proteins and restore ER protein homeostasis. When ER stress is irreversible, cellular functions deteriorate, often leading to cell death (Figure 2) [93]. There is mounting evidence that ER stress plays a significant role in the development and progression of varied diseases, including cancer and inflammation [93,94]. FaDOH-induced cell death is mediated via ER stress induction and the activation of the UPR.

Reducing the extent of ER stress by overexpressing the ER chaperone protein glucose-regulated protein 78 (GRP78) or by knocking down components of the UPR pathway decreased FaDOH-induced apoptosis. In contrast, raising the level of ER stress by inhibiting GRP78 enhanced the apoptosis triggered by FaDOH extracted from *Oplopanax horridus* (*O. horridus*) [95]. In addition, ER stress mediated panaxydol-induced apoptosis in MCF-7 cells [96]. Another study investigated the effect of a sub-toxic dose of 5 μM of FaDOH in a series of experiments and found that it increased the lipid content and number of lipid droplets (LDs) in human mesenchymal stem cells (hMSCs) and enhanced *PPARγ2* expression in human colon adenocarcinoma cells. The activation of *PPARγ* can enhance *ABCA1* expression [97]. FaDOH treatment showed an upregulation of *ABCA1* in colon neoplastic rat tissue, suggesting a function for this transporter in the redistribution of lipids and the enhanced creation of LDs in cancer cells, which may result in endoplasmic reticulum (ER) stress and cancer cell death [97].

## 4. Cancer

### 4.1. In Vitro

#### 4.1.1. Anti-Proliferative Activity

PAs derived from different plants exhibit potent cytotoxicity against a variety of cancer cells. These biologically active molecules engage directly or indirectly in biological processes, including cellular cycle arrest, HIF-1 (hypoxia-inducible factor-1 alpha) activation and signal transducer and transcriptional factor 3 (*STAT3*) suppression.

The anti-proliferative effects of FaOH isolated from carrots was initially shown in 2003. In addition, FaOH-type PAs show toxicity against human pancreatic carcinoma cells, but not against normal pancreatic cells, in vitro by modulating the expression of the genes involved in apoptosis, cell cycle, stress response and death receptors [42,98].

Treatment of leukaemia cell lines with carrot extract or isolated FaOH or FaDOH inhibited cell cycle progression, suggesting that carrots cause cell cycle arrest (G0/G1) in leukaemia cell lines [99]. Moreover, the cytotoxicity of FaOH, FaDOH and panaxydiol isolated from the dichloromethane extract of root celery was tested for its potential impact in a number of human cancer and leukaemia cell lines. All PAs examined exhibited moderate cytotoxicity against leukaemia, lymphoma and myeloma cell lines, although FaOH had significantly more activity against the lymphocytic leukaemia cells than FaDOH and panaxydiol [100]. In other studies, FaDOH also had less cytotoxic activity than FaOH and FaDOH3Ac [99,101].

#### 4.1.2. Pro-Apoptosis Activity

Cancer prevention and treatment depend on the use of a variety of bioactive compounds that inhibit the early stages of cellular transformation required for the development of the neoplastic phenotype, such as initiating autophagy, apoptosis or other forms of cell death such as oncosis or necrosis [102]. Apoptosis dysfunction is a major contributor to cancer development and progression. Tumour cells’ ability to avoid apoptosis can play a significant role in their resistance to traditional therapies.

One study investigated the effects of FaOH on human pancreatic ductal adenocarcinoma cell lines compared with normal pancreatic cells. FaOH regulated the genes related to pro-apoptosis, anti-apoptosis, apoptosis, cell cycle, stress and death receptors in adenocarcinoma cells more preferentially than in normal pancreatic cells [98]. FaOH suppressed pro-apoptosis genes (*BAD* and *HTRA-2*), anti-apoptosis genes (*Livin* and *XIAP*), a cell cycle controller (Phospho-p53 at amino acids serine 15, 46 and 392 (S15, S46 and S392), stress-related genes (*Clusterin* and *Hsp60*) and death receptor genes (*TNFR1* and *TNFSF1A*). In addition, FaOH increased cell cycle checkpoint phosphorylation (Phospho-Rad17 (S635)) and induced stress-related genes (*HO-1, HMXO1, HP32 and Hsp27*). Furthermore, FaOH-type and other PAs are potent inhibitors of pancreatic cancer cell proliferation [98].

Tumour recurrence and drug resistance are both facilitated by cancer stem-like cells (CSCs) [103]. *Hsp90* is known to enhance cancer cell survival and their ability to acquire anti-cancer drug resistance; its overexpression has been linked to a poor prognosis in human malignancies [104,105]. An in vivo study showed that orally administered FaOH significantly suppressed the proliferation of lung cancer in a mouse model without overt symptoms of toxicity at concentrations of 50 mg/kg body weight [103], which would correspond to a human dose of 4 mg/kg [106]. FaOH selectively inhibited carcinogenesis cells but not normal cells both in vitro and in vivo by inhibiting the function and viability of cancer stem-like cells of non-small-cell lung cancer by triggering apoptosis without enhancing *Hsp70* expression. Moreover, FaOH, at a low dose of 1 µM, induced apoptosis in cancer stem-like cells [103]. The pro-apoptotic function of panaxydol from *P. ginseng* was also tested on different cell lines to check whether the induction of apoptosis occurred preferentially in cancer cells. Indeed, panaxydol selectively induced apoptosis in malignant cancer cells [85].

#### 4.1.3. Gut Microbiota Composition

A study aimed to investigate whether the antibacterial effects of FaOH and FaDOH may be a mechanism of action in the antineoplastic properties of FaOH and FaDOH. They tested the effect of FaOH and FaDOH on gut microbiota composition in an AOM-induced rat colorectal cancer model. Rats treated with AOM were fed either a normal rat diet or a diet enriched with FaOH and FaDOH. Analysis of cecum faecal samples revealed a significant change in the gut microbiota among the groups. FaOH and FaDOH, which suppressed the growth of neoplastic tumours in the colon in a rat colon cancer model, modified the composition of low-abundance gut microbiota GM members, such as *Lactobacillus reuteri,* and high-prevalence *Turicibacter*, which was also correlated with a reduction in the number of macroscopic sites of neoplasms. Thus, this study demonstrated that modifications in the gut microbiota may play a significant role in the colon-protective action of FaOH and FaDOH against neoplastic transformation [107].

#### 4.1.4. Other Effects

FaOH stimulated the differentiation of primary mammalian cells at concentrations as low as 0.004 to 0.4 µM, whereas cytotoxic effects were observed at concentrations of >4 µM [108]. Moreover, one study evaluated PAs (FaOH and FaDOH) isolated from carrots in non-cancerous human intestinal epithelial cells (FHs 74 Int. cells) and intestinal cancer cells (CaCo-2). The growth–inhibition response was seen in concentrations above 1 μg/mL (~4 μM), with maximum inhibition at 20 μg/mL (~80 μM). The FaOH showed a higher inhibitory potency compared with FaDOH. In addition, cancer cells treated with combinations of FaOH and FaDOH showed a synergistic response for the inhibition of cell growth [109]. FaOH purified from carrots inhibited caspase-3 expression to prevent cell death and reduced basal DNA strand breakage in CaCo-2 cells. Thus, FaOH is either pro-survival or pro-death in a concentration-dependent manner in CaCo-2 cells. The effects of FaOH on CaCo-2 cells appear to be biphasic, with low and high concentrations of falcarinol inducing pro-proliferative and apoptotic characteristics, respectively [110].

PAs have other effects relevant for cancer. PAs can be used to heal or relieve symptoms by interacting with other foods or drugs. Cisplatin, which has nephrotoxicity as a side effect, is a therapeutic drug for various solid tumours. FaDOH attenuates cisplatin-induced injury and down-regulates mRNA levels of *TNF-α, IL-1β* and the protein expression of p-NF-κB p65 in mice [111].

Another study demonstrated the effects of FaOH, FaDOH, FaDOH3Ac and falcarindiol 3,8-diacetate on breast cancer multidrug resistance protein (BCRP/ABCG2), a xenobiotic efflux transporter that causes chemotherapy resistance in cancer. PAs inhibited mitoxantrone efflux (an ABCG2 substrate) in HEK293 cells overexpressing ABCG2. In a vesicular transport assay, a concentration-dependent inhibition of methotrexate (another ABCG2 substrate) uptake into ABCG2-overexpressing Sf9 insect cell membrane vesicles was observed. PAs also inhibited baseline and sulfasalazine-stimulated vanadate-sensitive ATPase activities in ABCG2-overexpressing Sf9 insect cell membrane vesicles. This study suggested that PAs might mitigate multidrug resistance in chemotherapy. As ABCG2 may play a role in the absorption and disposition of PAs, there may be food–drug interactions between PA-rich foods and ABCG2 substrate drugs [112].

### 4.2. In Vivo

There is not yet any direct in vivo evidence supporting an anti-cancer role of PA in humans. Observational human studies have reported that carrot consumption was associated with a reduced risk of several cancer types. For example, in a prospective cohort study including 57,053 Danes, an intake of 2–4 or more raw carrots each week (>32 g/day) was associated with a 17% reduction in the risk of colorectal cancer [16], pancreatic cancer and leukaemia [113] compared to individuals with no intake of raw carrots. One experimental study with carrot juice (500 mL) containing approximately 18 mg FaOH reduced the level of inflammatory cytokines *IL-1* and *IL-16* significantly in LPS-stimulated human blood an hour after intake compared with before the intake of carrot juice [114]. More detailed studies were performed using animal models. The consumption of carrot powder reduced the growth of intestinal tumours in an Apc^Min/+^ mouse colon cancer model [115,116]. A study examined colon preneoplastic lesions in AOM-treated rats that were fed carrots (10% freeze-dried carrot with a natural concentration of FaOH at 35 µg/g), FaOH (purified FaOH mixed at 3.5 µg/g in food) or a control for 18 weeks. The number and size of lesions decreased significantly in the rats that received either one of the two experimental treatments compared to the control group, indicating that carrots and FaOH slowed the growth of aberrant crypt foci (ACF) and tumours [117]. In a similar study, again using AOM-treated rats as a colon cancer model, feed containing a mixture of FaOH and FaDOH at concentrations four times higher than the previous study significantly reduced the number of tumours >1 mm, from 21 in controls to 12 in PA-treated rats [118]. An inverse correlation was found between a higher intake of a combination of FaOH and FaDOH with the multiplicity of colorectal neoplastic lesions [46]. These studies support the hypothesis that diets rich in FaOH and FaDOH can be a preventive treatment of colorectal cancer. A human dose of PAs corresponding to a 2017 rat experiment would be 24 mg per day for a 70 kg person, which could be provided by consuming 260 g per day of the cultivated carrot cultivar ‘Nantes Empire’ [118].

PAs from ginseng have shown selective tumour reduction activities similar to chemotherapeutic agents. Panaxydol isolated from *Panax ginseng* (*P. ginseng*) inhibited tumour growth in mouse tumour models, including PC3 human prostate cancer xenograft and mouse renal carcinoma (Renca) cells. BALB/c nude mice bearing PC3 or Renca cell tumours were injected with panaxydol every two days for a course of three weeks. Panaxydol inhibited the growth of the PC3 xenograft dose-dependently, with complete suppression at 100 mg/kg. Panaxydol also reduced Renca tumour size in dose-dependent manner, demonstrating an in vivo anticancer effect in this model [96].

## 5. Polyacetylene Toxicology and Pharmacokinetics

### 5.1. Toxicology of PAs

PAs in high concentrations have toxic effects that depend on cell sensitivity. In in vitro studies, FaOH has shown cytotoxic activity against intestinal cell lines at concentrations of 4 µM [108] and 10 μM [109]. In addition, FaDOH has shown a toxic activity in human colon adenocarcinoma (HT-29) cells in concentrations >50 µM, while it exhibited a toxic effect on human mesenchymal stem (hMSC) cells in concentrations > 20 µM [97]. In another study, FaDOH and panaxydol showed toxicity at concentrations of 40 μM [109]. In vivo, FaOH showed neurotoxic effects at a high concentration (LD50 = 100 mg/kg) when injected into mice [119], whereas FaDOH had no neurotoxic effects in rats when injected with similar concentrations (LD50 > 200 mg/kg) [120]. However, by inhibiting the Foxo–Notch axis, FaDOH can disrupt the maintenance of normal neural stem cells and modify the balance between the self-renewal and differentiation of neural stem cells with negative consequences [121]. There was also a report on the modulation of GABA_A_ receptors by FaOH, which may underlie a sedative effect [122]. Mammals have not been observed to be poisoned after the voluntary consumption of FaOH-type PAs in natural sources; this is probably related with the bitter taste of PAs, in particular FaDOH, which causes a bitter or burning sensation when occurring in concentrations > 40 µM [123], thus preventing the eating of unsafe amounts of vegetables with too high levels of these PAs. This contrasts with other types of polyacetylenes such as oenanthotoxin, which is found in the neurotoxic plant hemlock water dropwort [124]. However, while FaOH-type PAs can also cause neurotoxic symptoms, this requires much higher concentrations than those that occur in edible plants, and therefore their presence in food plants is deemed harmless [125]. FaOH from ivy (*Hedera helix* L.) has a moderate allergic potential on human skin. Repeated direct contact with ivy or other plants containing FaOH can cause sensitization in susceptible individuals and subsequently lead to allergic contact dermatitis after long-term frequent exposure to skin [126]. Most patients with this rare condition become sensitised in occupational settings, e.g., plant nursery workers handling ornamental plants [127]; only very few cases of PA-related contact dermatitis from Apiaceous vegetables have been reported [128]. This contact dermatitis may be related to falcarinol selectively alkylating the anandamide binding site in the CB1 receptor [129].

As far as we are aware, no research has been published on the safety and side effects of falcarinol-type PAs in other contexts than food safety or contact dermatitis. Therefore, it is necessary to carry out research on the toxicological evaluation and potential toxicity mechanisms of PAs and to establish scientifically justified safe doses and applications as a prerequisite for their use in therapeutic and clinical applications.

### 5.2. Pharmacokinetics of PAs

The study of pharmacokinetics determines the fate of drugs supplied externally to a living body [42]. Pharmacokinetic research, such as pharmacodynamic and toxicological research, has become an essential part of drug preclinical and clinical research. It is critical in the development of new drugs, the improvement of dosage forms and the study of dosage form mechanisms [130].

Two studies have reported on the bioavailability of serum FaOH concentrations in humans after the consumption of carrot juice. One study used three doses of fresh carrot juice providing 19, 33 or 49 µmol FaOH and demonstrated the dose-dependent bioavailability of this compound, reaching 0.010 µM (4.0 ng/mL) after the highest dose [19]. A recent study prepared juice from 30 g of freeze-dried carrot powder containing approximately 18 mg FaOH, corresponding to 300 g raw carrot (two–three carrots). The powder was mixed with water to a total of 500 mL and given to participants. Serum FaOH concentrations reached their peak at 1 h after consumption, and the peak concentration was 0.9–4.0 ng/mL. FaOH had a half-life of 1.5 h in human serum [131]. Another study reported measuring the pharmacokinetics of FaOH in mice. FaOH was intravenously (IV) administered to mice at 5 mg/kg and orally administrated at 20 mg/kg to determine the pharmacokinetic parameters in the plasma and tissue using LC-MS/MS. FaOH reached a peak plasma concentration of 8.24 μg/mL after IV administration and then declined in a multiphasic manner. A plasma analysis of FaOH after oral administration showed that the compound’s concentration quickly reached a maximum of 1.72 μg/mL in 1 h. The plasma concentration then decreased in a multiphasic manner, reaching a final measurable concentration of 32.2 ng/mL after 24 h. FaOH had a half-life of 1.5 h when IV injected and 5.9 h when administered orally, with a bioavailability of 50.4%. The mice did not show any toxicity up to 300 mg/kg orally [132]. When mice were orally given 20 mg/kg of FaOH, the FaOH concentrations in the colon tissue were the highest at 2 h after treatment at 121 ng/mL [132]. The very high murine values contrast remarkably with the similarity of the maximal plasma concentration ranges reported in the above two human studies. However, the lack of detail in the description of the LC-MS methodology in the mouse paper opens the possibility that this may reflect issues with the methodology rather than substantial differences in bioavailability between the species; additional bioavailability studies in mice would be useful to resolve this important question.

## 6. Conclusions

Food plants of the Apiaceae and Araliaceae families rich in PAs have important potential regarding cancer prevention. The findings reviewed here (Table 1) consistently support that PAs are anti-neoplastic natural phytochemicals with the potential for advancement into multiple applications in cancer prevention and treatment and as leading compounds in the discovery of new anticancer drugs. The mechanisms of the action attributed to PAs are similar to those of many other anticancer drugs, which include triggering cell cycle arrest, apoptosis, UPR and reducing inflammation but potentially with lower toxic side effects. The PA concentrations in widely consumed vegetables such as carrots are sufficiently high to potentially provide substantial chemo-preventive effects within the recommended vegetable and fruit intake of 400 g per day while at the same time being sufficiently low to exclude concerns about toxicity from these dietary sources. PAs have significant inhibitory effects on multiple cancer cell pathways, indicating anti-proliferative and anti-tumorigenic properties.

Nevertheless, much work still needs to be carried out to assess and possibly develop the medical development of PAs. Despite being widely distributed in plants, PAs are instable and present in relatively small amounts, which makes it challenging to isolate abundant polyacetylenes from natural sources for large-scale experiments. Improved isolation methods or the development of improved stereospecific chemical synthesis procedures will provide better feasibility for further study. Future studies are needed to determine the safe doses of PAs in humans, which is a prerequisite for any non-food application, whether as food supplements or drugs. In this regard, additional studies are needed on animals and humans for a better determination of the toxicological effects of PAs. Pharmacokinetic studies on PAs are limited, and the pharmacological actions of many PAs are unknown. Advancing this research will provide a scientific foundation for assessments of their potential for clinical applications and new drug development. NF-κB pathways are influenced by PAs, indicating their significance in terms of not only cancer prevention and treatment but also various other biological processes. However, future studies should focus on investigating the exact mechanism of action of PAs, particularly on NF-κB pathways, to distinguish the influence of PAs on gene transcription, translation or post-translation functions, such as enzymatic activity. Animal experiments, clinical studies and human intervention studies must be conducted to investigate and compare health benefits of PAs in whole foods or isolated PA metabolites using biomarkers indicating inflammation and other cancer-related processes to guide the optimisation of the implementation of affordable food-based cancer prevention programmes.

## Figures and Tables

**Figure 1 foods-12-01192-f001:**
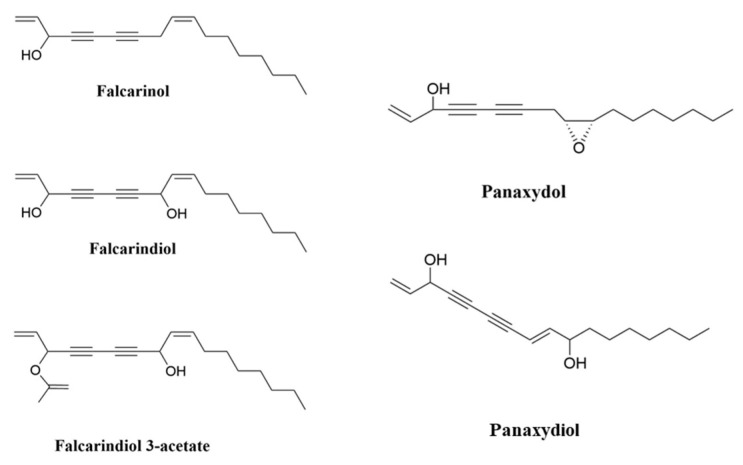
Chemical structures of FaOH, (3R)-falcarinol (also known as panaxynol); FaDOH, (3R,8S)-falcarindiol; FaDOH3Ac, (3R,8S)-falcarindiol-3-acetate; panaxydol; and panaxydiol.

**Figure 2 foods-12-01192-f002:**
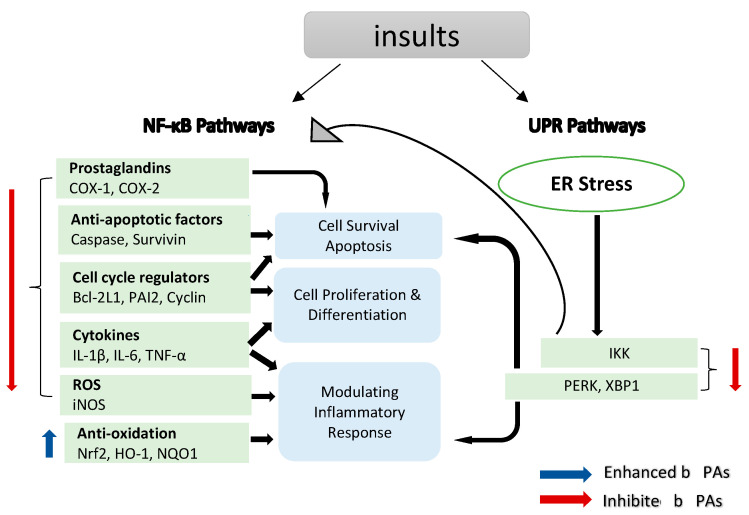
NF-κB target genes implicated in the onset and progression of inflammation. NF-κB is a transcription factor that is inducible. After activation, it can regulate inflammation by activating the transcription of several genes. NF-κB regulates cell proliferation, apoptosis and differentiation in addition to promoting the production of pro-inflammatory cytokines and chemokines. Endoplasmic reticulum (ER) stress results in an inflammatory unfolded protein response (UPR). Stress on the ER induces apoptosis by activating inflammation. This can be accomplished by stimulating IKK complex (element of the NF-κB) or (X-box binding protein 1) XBP1 and (protein kinase R-like ER kinase) PERK through a mediator. These trigger the release of pro-inflammatory molecules, hence accelerating cell death.

**Figure 3 foods-12-01192-f003:**
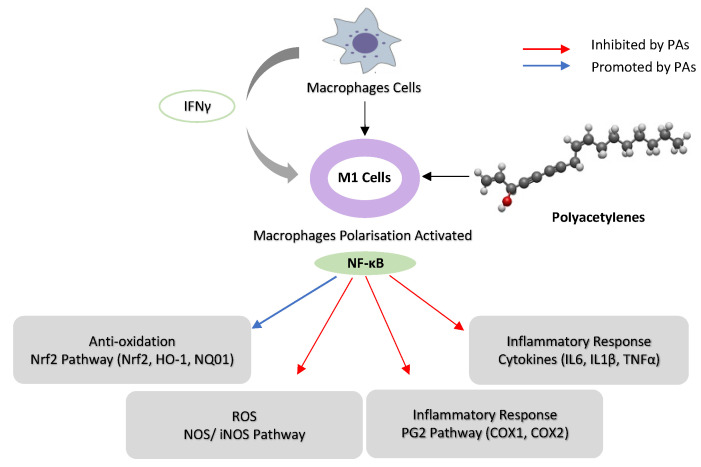
Schematic representation of the possible mechanism of immunoregulation activity of poly-acetylenes (PAs) in macrophages. (Interferon-γ) IFN-γ activates macrophage cells to M1, and PAs downregulate NF-κB activities in M1 macrophages by inhibiting *iNOS, COX-1* and *COX-2*. PAs suppress the inflammatory response by inhibiting cytokines (*IL-16, IL-1β* and *TNF-α*) and upregulating Nrf2 pathway (*HO-1* and *NQO1*) in macrophages. PAs structure modified from Mplanine (2022) https://www.wikiwand.com/en/Falcarinol#Media/File:Falcarinol_3D_BS.png (accessed on 1 January 2023).

**Table 1 foods-12-01192-t001:** Studies of polyacetylene compounds extracted from or contained in natural food and herbs and their applications, doses, time and effect or pathway assessed.

Compound/Source	Dose/Route	Time	Model	Effect/Pathway Investigated	Ref.
FaOH/carrot	Orally 10 mg/kg	1 weeks	Mice	Activate Nrf2 pathway	[5,92]
FaOH/*P. quinquefolius*	0.5–100 µM/mL	12 h	Primary Macrophages	Induce DNA damage/apoptosis	[8]
FaOH/*P. quinquefolius*	Orally 0.01–1 mg/kg	2 weeks	Mice	Reduce inflammation/Induce apoptosis	[8]
PAs/purple carrot	6.6 or 13.3 µg/mL	16 h	RAW 264.7 cells	Reduce secretion of the proinflammatory cytokines (IL-6, IL-1β, TNF-α)	[45]
FaDOH/*P. quinquefolius*	50 µM/mL	30–60 min	Astrocytes cells	Modulate NF-κB Pathway	[64]
FaOH/*S. divaricata*	Orally 0.1–1 mg/kg	1 weeks	Mice	Modulate NF-κB pathway/proinflammatory	[61]
FaOH/*S. divaricata*	1–10 µM/mL	3.5–24 h	Microglia cells	Reduce nitric oxide secretion/NF-κB pathway	[61]
FaOH, FaDOH/carrot	Orally 0.16–35 µg/g feed	20 weeks	Rats	Reduce tumour growth in colon/NF-κB pathway	[46]
FaOH/*H. moellendorffii*	6.25–50 µg/mL	20 h	RAW 264.7 cells	Activate ROS/Nrf2/HO-1 signaling pathway/Modulate NF-κB pathway	[65]
FaOH/*P. quinquefolius*	0.5 µM/mL	6 h	Macrophages cell lines	Activate Nrf2 pathway/NF-κB pathway	[85]
Panaxydol/*P. ginseng*	50 µg/mL	6 h	MCF-7 cells	Induce ROS generation, induce apoptosis/activate Nrf2 pathway	[88]
FaOH, FaDOH/carrot	6.25–50 µM/mL	24 h	Myotube cells	Modulate ROS pathway	[89]
FaDOH/*N. incisum*	2.5 µM/mL	24 h	HEK293 cells	Modulate Nrf2/ARE pathway	[90]
FaDOH/*O. horridus*	10 µM/mL	8 h	Colon cancer cells	Modulate ER/UPR pathway	[95]
Panaxydol/*P. ginseng*	20 µg/mL	2–4 h	MCF-7 cells	Modulate ER/EGFR pathway/induce apoptosis	[96]
FaDOH/carrot	5 µg/mL	24 h	HT-29/hMSCs cells	Modulate PPARγ pathway	[97]
FaOH/*O. horridus*	0.3 µg/mL	48 h	PANC-1 cells	Inhibit human pancreatic cancer cells/Induce apoptosis	[98]
FaOH, FaDOH/carrot	25–100 µM/mL	24 h	Leukaemia cells	Induced apoptosis in leukaemia cells/arrest of cell cycle	[99]
FaOH/*P. ginseng*	Orally 50–100 mg/kg mouse	8 weeks	Mice	Reduced lung tumorigenesis	[103]
FaOH/*P. ginseng*	1 µM/mL	-	NSCLC cells	Induce apoptosis	[103]
FaOH, FaDOH/carrot	Orally 7 µg/g feed	20 weeks	Rats	Alter gut microbiota, *Lactobacillus reuteri*, *Turicibacter*	[107]
FaOH/carrot	0.5–100 μM/mL	72 h	CaCo-2 Cell	Induce proliferation/apoptosis	[110]
FaDOH	IP injection/50–100 mg/kg	4 days	Mice	Activate Nrf2 pathway/NF-κB pathway	[111]
FaOH, FaDOH, FaDOH3Ac/carrot	20 µM/mL	105 min	HEK293 cells	Modulate ABCG2 pathway	[112]
FaOH in purple carrot juice	Orally approx. 18 mg in 500 mL	1 h	Human	Reduce secretion of the proinflammatory cytokines IL-1α and IL-16	[114]
PAs in carrot powder	Orally approx. 20 µg/g feed	12 weeks	Mice	Reduced tumours in intestine	[116]
PAs in carrot powder	Orally approx. 20 µg/g feed	10 weeks	Mice	Reduced tumours in intestine	[115]
Panaxydol/*P. ginseng*	IP injection/50–100 mg/kg	30 days	Mice	Reduced syngeneic and xenogeneic tumours	[96]
FaOH/carrot	Orally 3.5 μg/g feed	18 weeks	Rats	Reduced tumours in colon	[117]
FaOH+FaDOH/carrot	Orally 7 + 7 μg/g feed	18 weeks	Rats	Reduced tumours in colon	[118]

## Data Availability

Data are contained within the article.

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
