# Peer review of "Pathways Affected by Falcarinol-Type Polyacetylenes and Implications for Their Anti-Inflammatory Function and Potential in Cancer Chemoprevention"

_foods, 2023, doi:10.3390/foods12061192_

Round 1
Reviewer 1 Report
In the manuscript entitled “Mechanisms of Action of Polyacetylenes’ Anti-Inflammatory Mechanisms of Action of Polyacetylenes’ Anti-Inflammatory Treatment”, the authors reviewed the anticancer effects of Polyacetylenes’. This manuscript, in general, is interesting and well-written; however, I believe that it should be thoroughly modified and updated because of the following:
1. The authors are encouraged to propose another figure to better explain their message, focusing on multiple cell signaling pathways within the cancerous cells.
2. The authors should be more cautious in their conclusions because the efficacy of polyacetylenes in the clinical utility of tumors is all to demonstrate.
3. Authors should also discuss the limitations of using polyacetylenes in chemoprevention and their pharmacokinetics.
4. Some studies must be tabulated. Authors must include at least one table of relevant studies.
Author Response
Response to Reviewer 1 Comments
Point 1: The authors are encouraged to propose another figure to better explain their message, focusing on multiple cell signalling pathways within the cancerous cells.
Response 1: We thank you for your suggestion. Figure 2 has been improved and figure 3 has been added for a better explanation. However, the full mechanisms of PAs in inflammation and cancer are not yet fully understood. Therefore, the title has been changed to make it more clear what the message is.
Point 2: The authors should be more cautious in their conclusions because the efficacy of polyacetylenes in the clinical utility of tumours is all to demonstrate.
Response 2: Mentions of ‘treatment’, ‘proof’, and similar terms have been removed (even if stated by the author of the cited study), to emphasise that this is an area with active research, where many more studies are needed before certainty can be claimed. Similarly, where relevant the use of ‘anti-disease’ or ‘anti-symptom’ statements (such as ‘anti-cancer’ or ‘anti-inflammatory’) have been replaced with less definitive terms unless referring to a specifically measured outcome or to a generally accepted well-defined mechanism (e.g. anti-inflammatory function).
Point 3: Authors should also discuss the limitations of using polyacetylenes in chemoprevention and their pharmacokinetics.
Response 3: Pharmacokinetics effects are added to section 5.2. We agree that information about using PAs in chemoprevention and pharmacokinetics as well as specific assessments of PAs toxicity are still not fully understood. Please see the conclusion (section 6, lines 527-531) for more information added in this regard.
Point 4: Some studies must be tabulated. Authors must include at least one table of relevant studies.
Response 4: We thank you for your suggestion, and we agree that we should include a table containing different polyacetylene compounds and their applications, doses, time, and mechanism of action with their references that are included in the manuscript. Please, See Table 1, line 543.
Reviewer 2 Report
This review highlights the literature investigating the anti-inflammatory mechanisms of polyacetylenes and how they may be leveraged in cancer prevention and treatment.
Overall, the review is clear with very minor grammar and syntax issues. This review has the potential to provide a significant contribution to the field with some key major revisions. The review requires a more detailed interpretation of the studies, currently the results are well stated with very little interpretation beyond what is stated in the original publication. It is unclear what the strengths and limitations of these experiments are with regard to the overall body of literature. The review needs to consistently report the experimental conditions (polyacetylene source, dosing/timing, route of administration, model, etc.) for each study included in this review. We recommend this information be incorporated into a table with each primary manuscript reference included.
Additionally, some organizational adjustments will improve the overall flow. For example, the organization structure in section 2.1 is confusing. It is unclear why there is a section 2.1.1 without a 2.1.2. Section 2.1.1 doesn’t seem to be a subsection for the content in 2.1.
Remove the use of passive voice throughout (for example there is an excessive use of “has/have been shown”).
Limit the use of vague quantities, such as many, numerous, several, etc.
If an abbreviation is introduced and defined, use the abbreviation instead of the full word/phrase after the initial sentence. For example, line 33 introduces “polyacetylenes (PA)” but then this abbreviation is not used much in the rest of the text; Line 45. Line 45, falcarinol can be replaced by the abbreviation.
Specific comments:
Improvements to the figures:
• Figure 1: if applicable, add the abbreviations used in the text for each chemical, either in the figure or legend.
• Figure 2, does not currently support the text well. For example, line 190 includes a Figure 2 call-out in reference to NOS and iNOS, which are not included in the figure.
Line 33: characterize PA’s in more depth (structure, type of chemical, how many are known, metabolism, etc).
Lines 48-50, why was beta-carotene initially believed to be protective? Was this based on carrot intake? What effects (if any) did beta-carotene supplementation have on other chronic diseases besides lung cancer?
Line 55 clearly state what level of evidence supports that PA’s improved human health by stimulating anti-cancer and anti-inflammatory mechanisms.
Lines 59-60 why are purple carrots a promising research subject? This is unclear.
Line 68, reference 19- find more phytochemical-centered and up-to-date references.
Line 71, references 20- find more up-to-date reference, this was from 2007 and specific to China, does this adequately reflect the current drug market?
Line 75- reword: “known to modulate” this is a big claim and not established with the level of data in this review.
Line 84, references 22,23- need more recent citation for prevalence/incidence data.
Line 85, reference 24, update with a reference to the newest 2022 Hallmarks.
Section 2.1 is title Chronic inflammation Disease and Cancer, but the only cause of chronic inflammation discussed were bacterial or viral infections. Include background regarding metabolic disease and/or obesity.
Line 140, this seems redundant from what was stated in intro.
Line 141-142 how was NF-kB inhibited? Explain mechanism of action, or state the limitation if this is unknown.
Lines 173-176, add an interpretation as to why it was not chemopreventive but worked as a therapeutic, does this have to do with dosing, timing, administration, mechanism of action?
Section 3- are the anti-proliferative and pro-apoptotic effects seen in non-cancer control cells as well? Are these effects cancer cell specific or more like a chemotherapy and non-specific?
Section 398, have formal toxicology studies been performed in animals and people?
Author Response
Response to Reviewer 2 Comments
Point 1: The review requires a more detailed interpretation of the studies, currently, the results are well stated with very little interpretation beyond what is stated in the original publication. It is unclear what the strengths and limitations of these experiments are with regard to the overall body of literature. The review needs to consistently report the experimental conditions (polyacetylene source, dosing/timing, route of administration, model, etc.) for each study included in this review. We recommend this information be incorporated into a table with each primary manuscript reference included.
Response 1: We thank you for your suggestion, and we agree that we should include a table containing different polyacetylene compounds and their applications, doses, time, the effects and pathways investigated, with their references that are included in the manuscript. Please, see Table 1, line 543.
Point 2: Additionally, some organizational adjustments will improve the overall flow. For example, the organisation structure in section 2.1 is confusing. It is unclear why there is a section 2.1.1 without a 2.1.2. Section 2.1.1 doesn’t seem to be a subsection for the content in 2.1.
Response 2: The overall organization structure has been improved.
Point 3: Remove the use of passive voice throughout (for example there is an excessive use of “has/have been shown”).
Response 3: The overall wording has been improved.
Point 4: Limit the use of vague quantities, such as many, numerous, several, etc.
Response 4: Thanks for the suggestion. The texts have been revised according to your valuable suggestion.
Point 5: If an abbreviation is introduced and defined, use the abbreviation instead of the full word/phrase after the initial sentence. For example, line 33 introduces “polyacetylenes (PA)” but then this abbreviation is not used much in the rest of the text; Line 45. Line 45, falcarinol can be replaced by the abbreviation.
Response 5: We have replaced the words with their abbreviations.
Point 6: Figure 1: if applicable, add the abbreviations used in the text for each chemical, either in the figure or legend.
Response 6: Abbreviations have been included in the legend. Please, See line 72.
Point 7: Figure 2, does not currently support the text well. For example, line 190 includes a Figure 2 call-out in reference to NOS and iNOS, which are not included in the figure.
Response 7: iNOS/NOS are included in figure 2, however, figure 2 now is revised for improvement to better support the overall manuscript concept.
Point 8: Line 33: characterize PA’s in more depth (structure, type of chemical, how many are known, metabolism, etc).
Response 8: Thank you for mentioning this. We included some information about them. Please, see lines 47-53, in addition to the conclusion section 6, we mentioned that as a limitation.
Point 9: Lines 48-50, why was beta-carotene initially believed to be protective? Was this based on carrot intake? What effects (if any) did beta-carotene supplementation have on other chronic diseases besides lung cancer?
Response 9: The role of observational studies is specified, and one more reference added (assessing the effects of beta-carotene supplementation on mortality in general). This question is explained comprehensively in the Deding et al. 2020 reference, which therefore has been mentioned in this section as well. However, the manuscript's focus is on PAs, so the information about beta-carotene is intentionally brief, only included as background information on why the role of PAs was not investigated until after the beta-carotene hypothesis had been contradicted by subsequent experimental evidence.
Point 10: Line 55 clearly state what level of evidence supports that PA’s improved human health by stimulating anti-cancer and anti-inflammatory mechanisms.
Response 10: This paragraph has been improved with more relevant information added and moved to a more relevant section. Please, See lines 102-118.
Point 11: Lines 59-60 why are purple carrots a promising research subject? This is unclear.
Response 11: This information has been moved to the more relevant sections to improve the organisation of the manuscript. Please, See lines 108-118.
Point 12: Line 68, reference 19- find more phytochemical-centered and up-to-date references.
Response 12: This information has been moved to the more relevant sections to improve the organisation of the manuscript. We have also included later references. Please See line 65 (ref. 20-23).
Point 13: Line 71, references 20- find more up-to-date reference, this was from 2007 and specific to China, does this adequately reflect the current drug market?
Response 13: We thank you for pointing this out. Information and references have been updated to fit the general idea of this manuscript. Please, See lines 62-65.
Point 14: Line 75- reword: “known to modulate” this is a big claim and not established with the level of data in this review.
Response 14: We revised the wording to better represent the overall data. Please, See line 68.
Point 15: Line 84, references 22,23- need more recent citation for prevalence/incidence data.
Response 15: References have been updated. Please, See line 78 (ref. 25-28)
Point 16: Line 85, reference 24, update with a reference to the newest 2022 Hallmarks.
Response 16: The reference has been updated. Please, See line 79 (ref. 29)
Point 17: Section 2.1 is title Chronic inflammation Disease and Cancer, but the only cause of chronic inflammation discussed were bacterial or viral infections. Include background regarding metabolic disease and/or obesity.
Response 17: Sections have been adjusted to improve overall flow, as suggested. The main object of this manuscript is to present the latest findings regarding PAs and their impact on inflammation related to cancer and cancer in general. It is now well-established that chronic inflammation is a critical factor driving cancer development. Only a limited number of studies address this aspect, using DSS-induced colon inflammation and colon cancer development, both of which are discussed in this review (section 2.2). We agree that metabolic disease and obesity are important underlying causes of inflammation, however, there are no studies indicating that different causes for inflammation affect how PAs affect this inflammation. Therefore, the full inclusion of these aspects would deviate from the overall goal of this review.
Point 18: Line 140, this seems redundant from what was stated in intro.
Response 18: The sentence has been reformulated. Please, See lines 151-152.
Point 19: Line 141-142 how was NF-kB inhibited? Explain mechanism of action, or state the limitation if this is unknown.
Response 19: The exact mechanism of action of PAs on NF-kB inhibition remained unknown. We have clarified this now. Please, See lines 152-153.
Point 20: Lines 173-176, add an interpretation as to why it was not chemopreventive but worked as a therapeutic, does this have to do with dosing, timing, administration, mechanism of action?
Response 20: We appreciate your comment on this. Interpretation has been added. Please, See lines 184-194
Point 21: Section 3- are the anti-proliferative and pro-apoptotic effects seen in non-cancer control cells as well? Are these effects cancer cell specific or more like a chemotherapy and non-specific?
Response 21: In section 3 (section 4 in the revised manuscript), we have stated that in some studies non-cancer cells were used as control (See lines 337-346) and (lines 353-360). However, when it is not stated, it means that the data presented in that particular study have not included any non-cancer control cells, thus we can not make a statement in this regard.
Point 22: Section 398, have formal toxicology studies been performed in animals and people?
Response 22: There are no data reported on the toxicology of the PAs in animals or humans, primarily because these compounds are not commercially available from synthesis, and pure compounds isolated from plant material are prohibitively expensive. We have clarified this point in the revised manuscript.
Reviewer 3 Report
In this review paper, Alfurayhi et al summarized potential effects of polyacetylenes by targeting inflammation, oxidative stress and UPR pathway and highlighted their potential chemotherapeutic effects by preventing proliferation, promoting apoptosis and targeting gut microflora. Overall, the paper is well written.
Some comments:
1. The manuscript can be significantly improved if the authors can elaborate the effects of polyacetylenes on cancer initiation, promotion, progression and metastasis individually.
2. In the manuscript, the authors especially cited purple carrots and American Ginseng. If possible, please show any epidemiological evidence that reported these herbal medicines depicted anti-inflammatory and/or chemotherapeutic effects
Author Response
Response to Reviewer 3 Comments
Point 1: The manuscript can be significantly improved if the authors can elaborate the effects of polyacetylenes on cancer initiation, promotion, progression and metastasis individually.
Response 1: We thank you for your suggestion. However, the exact effects of PAs are still under discovery. The few available references with this kind of information are already included in the manuscript. We hope that the publication of the present review will help to inspire more research on these relations!.
Point 2: In the manuscript, the authors especially cited purple carrots and American Ginseng. If possible, please show any epidemiological evidence that reported these herbal medicines depicted anti-inflammatory and/or chemotherapeutic effects.
Response 2: The main object of this manuscript is to highlight and present the latest findings regarding PAs derived from different plants, not the plants themselves in regard to their mechanism of inflammation related to cancer and cancer in general. Regarding carrots, our colleague Charles Ojobor has carried out a meta-analysis of epidemiological studies of carrots and cancer, which is presently under review, see pre-preprint: Ojobor et al. (2023) Carrot Intake is Consistently Negatively Associated with Cancer Incidence: A Systematic Review and Meta-Analysis of Prospective Observational Studies. Available at SSRN: http://dx.doi.org/10.2139/ssrn.4316849. However, since the epidemiology does not directly provide any mechanistic information, we did not find this information so essential that it would be appropriate to cite an unpublished manuscript.
Reviewer 4 Report
The review article is interesting. The following recommendations are suggested
Abstract & Keywords. OK
Introduction. OK
Main body. Authors must specify when they refer to the inhibition or activation of enzymatic activity, decreased or augmented protein synthesis or overregulation or down regulation of transcripts involved in the mechanisms described por polyacetylenes. Title of section 2 must include also oxidative stress. Authors must further elaborate on the section 4 regarding possible toxic effects of polyacetylene.
Tables & Figures. Authors must include tables that summarize the in vitro and in vivo studies described in the article body, including the source, doses and duration, study model, major findings and references. This tables will allow readers to rapidly understand the main results included in the review paper. In addition, a figure including the integration of all mechanisms discussed in the article must be included.
Conclusions. OK
Author Response
Response to Reviewer 4 Comments
Point 1: Main body. Authors must specify when they refer to the inhibition or activation of enzymatic activity, decreased or augmented protein synthesis or overregulation or down regulation of transcripts involved in the mechanisms described por polyacetylenes. Title of section 2 must include also oxidative stress. Authors must further elaborate on the section 4 regarding possible toxic effects of polyacetylene.
Response 1: We thank you for your suggestion. Oxidative stress has been added to section 2, please see section 2.3, line 200. We have further elaborated on the toxicity of the falcarinol-type PAs, although mostly by specifying which aspects are still not fully understood. There are not enough references in this regard, particularly regarding the longer-term toxicity on mice and any data on humans.
Point 2: Tables & Figures. Authors must include tables that summarize the in vitro and in vivo studies described in the article body, including the source, doses and duration, study model, major findings and references. This tables will allow readers to rapidly understand the main results included in the review paper. In addition, a figure including the integration of all mechanisms discussed in the article must be included.
Response 2: We thank you for your suggestion, and we agree that we should include a table containing different polyacetylene compounds and their applications, doses, time, and mechanism of action with their references included in the manuscript. Please, see Table 1, line 543. Moreover, Figure 2 has been improved and figure 3 has been added for a better explanation. We have also changed the title of the review to specify our realistic level of ambition in this regard.
Round 2
Reviewer 1 Report
I found the revised paper quite improved. Indeed, the authors properly addressed the queries raised by the reviewers and added suggested figures and tables. This paper needs a few minor modifications.
I do suggest improving the following:
Rewrite the symbols properly throughout the manuscript-
Line 58- Figure 1 replace with Figure 1.
Line 200- In vitro and in vivo replace with In-vitro and in-vivo
Line 283- NF-kB replace with NF- ƙB
Line 401- COX1, COX2 replace with COX-1 COX-2
Line 620 - In vivo replace with In-vivo
Line 622- 18mg replace with 18 mg
Line 653 - IL1, IL16 replace with IL-1, IL-16
Line 669- 24mg replace with 24 mg
Line 670- 260g replace with 260 g
Line 685- 50μM replace with 50 μM
Line 686- 20μM replace with 20 μM
Line 687- 40μM replace with 40 μM
Line 697-40μM replace with 40 μM
Line 865- 400g replace with 400 g
Author Response
Thanks for the suggestion. The texts have been revised according to your valuable suggestion.
Reviewer 4 Report
Authors improved the quality of the manuscript as suggested by the reviewers.
Author Response
Thank you. The texts have been revised according to your valuable suggestion.